# Quality of routine health facility data for monitoring maternal, newborn and child health indicators: A desk review of DHIS2 data in Lumbini Province, Nepal

Keshab Sanjel[1]*, Shiv Lal Sharma[2], Swadesh Gurung[1], Man Bahadur Oli[3], Samikshya Singh[1], Tuk Prasad Pokhrel[4]

1 Abt Associates Inc., Kathmandu, Nepal, 2 Management Division, Department of Health Services, IHIMS Section, Ministry of Health and Population, Kathmandu, Nepal, 3 Health Directorate, Ministry of Health, Lumbini Province, Nepal, 4 Health Office Palpa, Ministry of Health, Lumbini Province, Nepal

* keshabsanjel@gmail.com

**Data Availability Statement:** All relevant data are within the manuscript.

## Abstract

### Introduction

Health-facility data serves as a primary source for monitoring service provision and guiding the attainment of health targets. District Health Information Software (DHIS2) is a free open software predominantly used in low and middle-income countries to manage the facility-based data and monitor program wise service delivery. Evidence suggests the lack of quality in the routine maternal and child health information, however there is no robust analysis to evaluate the extent of its inaccuracy. We aim to bridge this gap by accessing the quality of DHIS2 data reported by health facilities to monitor priority maternal, newborn and child health indicators in Lumbini Province, Nepal.

### Methods

A facility-based descriptive study design involving desk review of Maternal, Neonatal and Child Health (MNCH) data was used. In 2021/22, DHIS2 contained a total of 12873 reports in safe motherhood, 12182 reports in immunization, 12673 reports in nutrition and 12568 reports in IMNCI program in Lumbini Province. Of those, monthly aggregated DHIS2 data were downloaded at one time and included 23 priority maternal and child health related data items. Of these 23 items, nine were chosen to assess consistency over time and identify outliers in reference years. Twelve items were selected to examine consistency between related data, while five items were chosen to assess the external consistency of coverage rates. We reviewed the completeness, timeliness and consistency of these data items and considered the prospects for improvement.

### Results

The overall completeness of facility reporting was found within 98% to 100% while timeliness of facility reporting ranged from 94% to 96% in each Maternal, Newborn and Child

**Funding:** The author(s) received no specific funding for this work.

**Competing interests:** The authors have declared that no competing interests exist.

Health (MNCH) datasets. DHIS2 reported data for all 9 MNCH data items are consistent over time in 4 of 12 districts as all the selected data items are within ±33% difference from the provincial ratio. Of the eight MNCH data items assessed, four districts reported ≥5% monthly values that were moderate outliers in a reference year with no extreme outliers in any districts. Consistency between six-pairs of data items that are expected to show similar patterns are compared and found that three pairs are within ±10% of each other in all 12 districts. Comparison between the coverage rates of selected tracer indicators fall within ±33% of the DHS survey result.

## Conclusion

Given the WHO data quality guidance and national benchmark, facilities in the Lumbini province well maintained the completeness and timeliness of MNCH datasets. Nevertheless, there is room for improvement in maintaining consistency over time, plausibility and predicted relationship of reported data. Encouraging the promotion of data review through the data management committee, strengthening the system inbuilt data validation mechanism in DHIS2, and promoting routine data quality assessment systems should be greatly encouraged.

## Introduction

Routine health information system comprises data collection, analysis, dissemination and use that provides information at regular intervals and that is produced through routine mechanisms [1]. It aims to improve health management through optimal informational support [2]. Informational support is required for all levels of health management for planning, policy making, operational management and continuous quality improvement [3]. The functionality of the health system and the ability of policymakers to assess the impact of health system initiatives on population health heavily rely on the quality of routine health information generated in the health facilities [4].

Many countries, particularly in low-income settings, lack well-functioning information systems that can support health system strengthening. The large variety and volume of data produced in public health facilities through routine health systems are often overlooked due to their poor quality [5–7]. Specific data quality issues may arise, including incomplete, inconsistent, and irrelevant data, as well as imprecise estimates of the target population for coverage. These issues could limit the usefulness of the data for decision-makers [8, 9]. It is common to observe discrepancies between the findings derived from data generated in routine health information systems and those obtained through population-based surveys [10].

The Countdown to 2030, Sustainable Development Goals (SDGs) and other global initiatives emphasize the importance of routine health information system to monitor and measure progress and take corrective action [11–13]. Nepal is also the part of these initiatives and developed action plans to reduce preventable deaths for mothers and children and has made considerable investment in strengthening information systems, to support performance management and service delivery [14, 15].

The DHIS2 is customizable free open source software currently used in over 75+ countries to manage and visualize routine health information. It has advance system for transmission and aggregation of data faster than paper-based information systems [13, 16]. The Ministry of

Health and Population (MoHP) in Nepal introduced DHIS2 nationally as an electronic platform for data management since 2016. The DHIS2 Platform is being used as a national database for electronic management of the Health Management Information System (HMIS) data. Being a digital platform, it increases the accessibility of facility data for program managers at federal, provincial and local levels. Although the DHIS2 is operational in Nepal, few assessments suggest the health facilities usually report incomplete, untimely, incorrect and inconsistent data in HMIS that do not provide a good basis for knowledge-based decision-making on health system [17, 18]. Reported data might be subjected to data quality limitations like presence of measurement error, missing values and human errors in data collection, tally, data entry and calculation [17, 18]. This might jeopardize the efforts in achieving targets both at subnational and national levels.

Data quality monitoring could be done to figure-out how much confidence we can place in the routine data that are used to measure performance and to articulate relative strengths and weaknesses of the routine data sources. Therefore, this study aims to assess the quality of facility based MNCH data in DHIS2 to monitor priority maternal and child health indicators in Lumbini Province, Nepal. The key maternal and child health indicators of Lumbini province are reported to be either above or consistent with national average but challenge remains to achieve national as well as SDG targets. Data reveals that, almost 84% of births in the province are delivered at health facilities and 87% of births are assisted by a skilled birth attendants [19]. Almost 97% of children are fully immunized, incidence of diarrhoea and pneumonia among under-five children (per 1000) are 346.6 and 29.3, respectively [20]. Still, 25% of children under age 5 are stunted and 29% are underweight [19]. The assessment will review the quality of key indicators reported through the routine information system in Lumbini Province.

## Methods

### Study Setting

Lumbini Province is one of the seven provinces of Nepal. It is located within western region and comprises a total population of 5.1 million [21]. The province has 12 districts and 109 local levels. Provincial Ministry of Health (MoH) oversees the health service delivery in secondary and tertiary facilities, and health sections in each local levels oversee the basic health facilities including basic hospitals, health posts, urban health clinics, basic health service centers and others.

A total of 1047 health facilities across 12 districts reported to DHIS2 in 2021/22. In general, health facility staff complete the Health Management Information System (HMIS) registers to document the services they provide each day. Every month, selected data in these registers are tallied, compiled and summarized in paper based monthly HMIS reports and health facilities either enter into DHIS2 by themselves (72%) or send to the municipal health section (28%) for data entry into DHIS2.

### Study design and data sources

We employed a facility-based descriptive study design to examine the facility-based MNCH service data in DHIS2. In 2021/22, DHIS2 contained a total of 12,509 reports. Reports refer to the monthly report submitted by each health facility through HMIS system (Report code: HMIS 9.3/9.4/9.5). This report includes individual sections for immunization, safe motherhood, nutrition, and IMNCI. Of those reports, monthly aggregated DHIS2 data for the reference year 2021/22 were downloaded at one time and included priority maternal and child health related data elements (Table 1). Additionally, we downloaded data for three previous fiscal years (FY 2018/19 to 2020/21) as comparison years for assessing the consistency of

**Table 1. Priority maternal, newborn and child health data items for data consistency review.**

| Priority maternal and child health data items | Data quality metrics | | | |
|---|---|---|---|---|
| | Consistency over time | Outliers in reference year (2021/22) | Consistency between related data | Consistency between DHS survey and DHIS2 data |
| **Data elements (in number)** | | | | |
| First Antenatal care (ANC) visit (any time) | | | × | |
| Pregnant women receiving deworming tablets | | | × | |
| Four ANC visits as per protocol | × | × | × | |
| Pregnant women receiving 180 Iron tablets | | | × | |
| Total institutional deliveries | × | × | × | |
| Total delivery presentations (Cephalic, Shoulder, Breech) | | | × | |
| Women received delivery incentive on transportation | × | × | | |
| 3 Postnatal Care (PNC) Visits as per Protocol | × | × | | |
| Children Immunized with BCG | × | × | | |
| Children Immunized with DPT-HepB-Hib 1st | | | × | |
| Children Immunized with PCV- 1st | | | × | |
| Children Immunized with Measles/Rubella-2nd | × | × | | |
| New Growth Monitoring visits (0–11 months) | × | × | | |
| Children with exclusive Breastfeeding practice | × | × | | |
| Diarrhoea cases (2–59 months) | | | × | |
| Diraahoea cases treated with ORS and Zinc | × | × | × | |
| Pneumonia cases (2–59 months) | | | × | |
| Pneumonia cases treated with antibiotics | | | × | |
| **Indicators** | | | | |
| Percentage of four or more ANC visits | | | | × |
| Percentage of institutional delivery | | | | × |
| Births assisted by skilled provider | | | | × |
| BCG coverage | | | | × |
| Measles Rubella1 | | | | × |

The symbol '×' denotes the selection of data items in each row corresponding to the data quality metrics mentioned in the respective column of the table above

reported data over time. We also accessed the Nepal Demographic and Health Survey (NDHS) 2022 key indicators report to measure the consistency of DHIS2 data with the estimates from external sources of data. For this, five tracer indicators are taken from both DHIS2 and NDHS survey and compared the results.

## Selection of priority maternal and child health indicators

DHIS2 listed four MNCH datasets: Immunization, Community Based Management of Neonatal and Childhood Illness (CBIMNCI), Nutrition, and Safe Motherhood, are considered for assessing completeness and timeliness of facility reporting. To select these dataset related data items, we selected priority MNCH related data items from the Data quality Review Toolkit (Module 1) developed by World Health Organization [22]. The selection of data items takes

into account the MNCH continuum of care approach, their ability to cross-verify and measure data quality issues, as well as their importance in program monitoring and evaluation, as stated in national and provincial policy documents [23–25]. (Table 1).

## Data analysis

We assessed the timeliness and completeness of facility reporting for the MNCH-specific datasets, as specified in DHIS2. We reviewed the consistency of the DHIS2 data according to metrics outlined by the World Health Organization data quality report card and a toolkit for facility data quality assessment [22, 26]. Data items are also contextualized based on the data validation rules set in the DHIS2 platform and considered the data review guideline practiced by the Lumbini Province (Table 2). We applied each of the metrics to assess the data quality status at the district level.

The analysis of mentioned dimensions was performed using a Microsoft Excel Worksheet. Descriptive analyses, including frequency, percentage, mean, and ratio, were calculated to assess the quality of MNCH data items using the measurements explained in Table 2. The analysis focused on completeness and timeliness of health facility reporting, consistency over time, identification of outliers in the current year, consistency between related data, and external consistency of the data.

**Table 2. Data quality dimensions, metrics and benchmarks.**

| Data Quality metric | Measurement/analysis |
|---|---|
| **Dimension 1: Completeness and timeliness** | |
| **Completeness of facility reporting** | Percentage of expected monthly reports of MNCH datasets submitted in 2021/22 |
| Extent to which each health facilities submitted monthly reports in DHIS2 (calculated based on MNCH datasets) | *Completeness of facility reporting should be 100%* |
| **Timeliness of facility reporting** | Percentage of expected monthly reports of MNCH datasets submitted on time (monthly reporting by 14<sup>th</sup> of next month) in 2021/22 |
| Extent to which each health facilities submitted monthly reports on time (calculated based on MNCH datasets) | *Timeliness of facility reporting should be 90% or higher* |
| **Dimension 2: Internal consistency** | |
| **Consistency over time** | Ratio of value of indicator for reference year (2021/22) to the mean of preceding 3 years (FY 2018/19 to 2020/21) |
| The consistency of the values for key MNCH indicators in the most recent year compared with the mean value of the same indicator for the previous three years combined | *Ratio of value of indicator for reference year should be within ±33% of mean of preceding 3 years* |
| | [If any district has a difference by more than ±33% in more than one indicator, it is also counted once as it is the same district] |
| **Outliers in the current year** | Number of moderate outliers (±2SD from the mean) and extreme outliers (±3SD from the mean) of monthly values during the reference year |
| Extent to which the values reported for a given indicators are extreme and potentially implausible | *Value of indicator should be within ±2SD from the mean* |
| **Consistency between related data** | Ratio for values of indicator-pairs (a set of related data items) that have a predictable relationship |
| Extent to which the values for two or more indicators exhibit the predicted relationship and how much the data is trustworthy | *Indicator-pairs that should be roughly equal should be within ±10% of each other (i.e., no extreme difference)* |
| *Dimension 3: External consistency of data* | |
| Consistency between household surveys and reported data in DHIS2: Extent to which values for given indicators agree with an external data source- Demographic and Health survey | Ratio of indicator values in most recent household survey for facility catchment areas to matching facilities in DHIS2 *Indicator values from facility reports in DHIS2 should be within ±33% of household survey value or within confidence limits of household survey.* |

## Results

### Completeness of maternal, newborn and child health datasets

All the health facilities need to complete the assigned datasets each month for reporting completeness. At the province level, the completeness of each MNCH datasets in 2021/22 ranged from 99.7% to 99.9%. At the district level, 10 out of 12 districts have 100% completeness for immunization dataset, 7 out of 12 districts have 100% completeness for nutrition, and IMNCI datasets (in each) and 8 out of 12 districts have 100% completeness for safe motherhood datasets. Overall, The MNCH datasets in five districts (Rolpa, Gulmi, Arghakhanchi, Dang, and Bardiya) have achieved 100% completeness. Very few of the health facilities have not completed the assigned datasets in the reference year (Fig 1).

### Timeliness of maternal, newborn and child health datasets

This dimension is assessed by measuring whether the health facilities which submitted reports did so before a predefined deadline set by government. By district, 9 out of 12 districts have more than 90% reporting rate on time for each of the selected datasets. The reporting timeliness of Rolpa, Pyuthan, and Bardiya districts is better in comparison to other districts. The timeliness of MNCH datasets in 2021/22 ranged from 94–96% considering provincial average (Fig 2).

### Internal consistency of reported data

This metrics examined the coherence between the same data items at different points in time (monthly, annual) and coherence between related data items. Major causes of inconsistency are an error during data entry, for example, when data are maintained in service registers, compiled in a tally sheet, and transferred to monthly reports and entered from a paper-based reports into DHIS2. Key dimensions included for assessing internal consistency include:

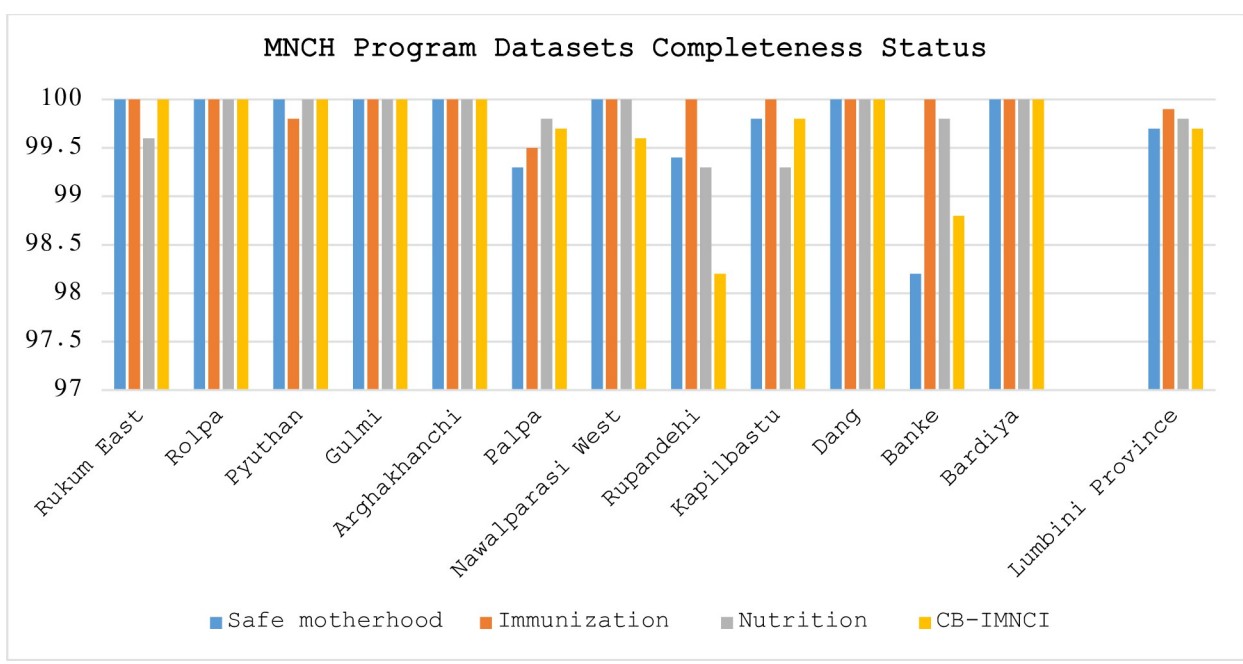

**Fig 1. Maternal, Newborn and Child Health (MNCH) program datasets completeness status.**

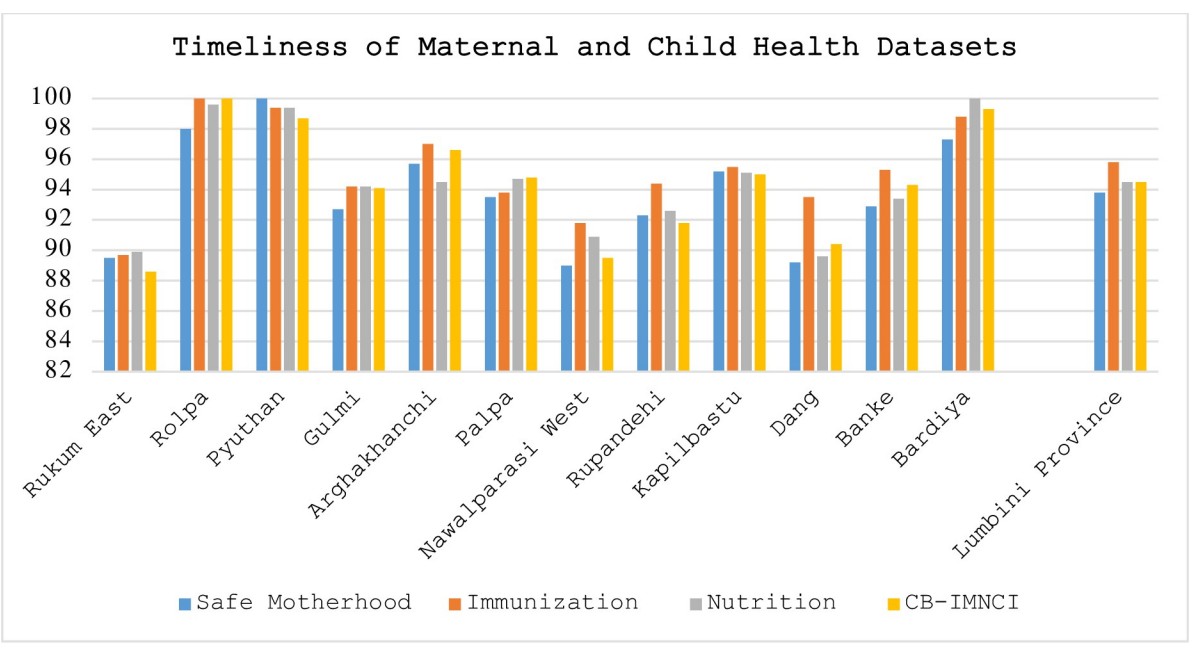

**Fig 2. Timeliness of maternal and child health datasets.**

## Consistency over time

The consistency of the time was assessed to observe whether the differences in values are expected from one year to the next. In the case where there is an existence of larger difference, it suggests the need for further scrutiny. While large differences usually suggest some type of reporting error, it is also possible that the introduction of a new intervention might have contributed to a significant percentage increase in indicator values from one year to the next. Consistency of the mean values is an indicator of reliability-meaning the greater probability that data source is trustworthy. This perspective examined the plausibility of reported data for 9 MNCH data elements in terms of the trends of reporting and determines whether reported values are extreme in relation to other values reported over four years. Table 3 depicts the consistency of the values in 2021/22 compared with the mean value of the same data item for the preceding three years combined. Eight out of 12 districts have a ratio that is more than ±33% difference from the provincial ratio in at least one data item. The data items with more than ±33% difference included: women who received transportation incentive (Palpa and Rupandehi), PNC visit as per protocol (Arghakhanchi and Bardiya), new growth monitoring (Dang) and exclusive breastfeeding (Rukum East, Palpa, Dang and Banke) (Table 3).

## Accuracy of event reporting: outliers in the reference year

This dimension assesses if the reported data over a period (monthly) follow a pattern with no significant variations. To achieve this, we initially conducted a normality test to identify the distribution of the data. Since the data exhibited a symmetrical distribution, we then examined the presence of outliers for eight maternal and child health data items in the reference year 2021/22. District wise, four out of 12 (Rukum East, Rolpa, Nawalparasi and Dang) have 5% or more of reported values across eight items are moderate outliers. However, extreme outliers were not reported in the reference year (Table 4). The outlier details for maternal and child health indicators are provided in the S1 and S2 Tables.

**Table 3. Consistency over time for priority maternal and child health indicators in DHIS2.**

| Tracer MNCH data elements | | Rukum East | Rolpa | Pyuthan | Gulmi | Arghakhanchi | Palpa | Nawalparasi West | Rupandehi | Kapilbastu | Dang | Banke | Bardiya | Province |
|---|---|---|---|---|---|---|---|---|---|---|---|---|---|---|
| Four ANC visit as per protocol | Ratio of Year 4 to mean of Year 1–3 | 1.2 | 1.0 | 1.0 | 0.9 | 1.1 | 0.9 | 1.1 | 1.3 | 1.2 | 1.0 | 1.1 | 0.9 | **1.1** |
| | ≥±33% difference from provincial ratio | 8.8 | -7.7 | -13.1 | -15.6 | -5.3 | -15.7 | -2.9 | 18.0 | 6.2 | -7.1 | 3.2 | -18.6 | |
| Institutional delivery | Ratio of Year 4 to mean of Year 1–3 | 1.2 | 0.9 | 1.0 | 1.0 | 1.0 | 0.8 | 1.0 | 1.0 | 1.2 | 0.9 | 1.1 | 0.9 | **1.0** |
| | ≥±33% difference from provincial ratio | 14.0 | -6.7 | -3.4 | -2.6 | 2.4 | -18.4 | 0.0 | 2.6 | 17.5 | -7.9 | 4.4 | -16.0 | |
| Women received delivery incentive on transportation | Ratio of Year 4 to mean of Year 1–3 | 1.2 | 1.0 | 1.0 | 0.8 | 1.2 | 0.4 | 1.1 | 1.5 | 1.3 | 0.8 | 1.2 | 0.9 | **1.1** |
| | ≥±33% difference from provincial ratio | 9.0 | -13.6 | -12.9 | -29.5 | 5.0 | **-66.9** | -5.8 | **35.5** | 12.5 | -28.6 | 3.4 | -19.4 | |
| PNC visit as per protocol | Ratio of Year 4 to mean of Year 1–3 | 1.7 | 1.7 | 1.6 | 2.0 | 3.8 | 1.9 | 2.7 | 2.1 | 2.4 | 2.2 | 2.0 | 1.2 | **2.0** |
| | ≥±33% difference from provincial ratio | -16.5 | -14.1 | -18.8 | -0.4 | **89.2** | -4.7 | **36.8** | 5.7 | 22.9 | 8.7 | 3.2 | **-39.0** | |
| Children immunized with BCG | Ratio of Year 4 to mean of Year 1–3 | 0.9 | 0.8 | 1.0 | 0.8 | 0.9 | 1.1 | 0.9 | 1.0 | 1.0 | 0.9 | 1.0 | 0.9 | **1.0** |
| | ≥±33% difference from provincial ratio | -9.0 | -16.1 | 1.3 | -10.9 | -9.2 | 20.5 | -4.2 | 0.4 | 5.6 | -2.4 | 7.3 | -6.5 | |
| MR2 coverge | Ratio of Year 4 to mean of Year 1–3 | 1.0 | 1.0 | 1.0 | 1.0 | 1.0 | 1.0 | 1.0 | 1.1 | 1.2 | 1.0 | 1.1 | 1.0 | **1.1** |
| | ≥±33% difference from provincial ratio | -9.2 | -4.7 | -2.6 | -7.0 | -9.7 | -9.2 | -4.3 | 0.8 | 15.1 | -0.5 | 0.3 | -2.1 | |
| Diraahoea treated with ORS and Zinc | Ratio of Year 4 to mean of Year 1–3 | 0.9 | 0.8 | 0.7 | 0.8 | 0.9 | 0.7 | 0.8 | 0.8 | 0.8 | 0.8 | 0.8 | 0.9 | **0.8** |
| | ≥±33% difference from provincial ratio | 18.0 | 1.8 | -10.7 | -5.3 | 14.1 | -15.8 | -4.6 | 3.4 | -1.4 | -1.1 | 3.8 | 10.6 | |
| New growth monitoring visit (0–11) | Ratio of Year 4 to mean of Year 1–3 | 0.7 | 0.9 | 0.8 | 0.8 | 0.8 | 0.7 | 0.9 | 1.2 | 1.1 | 1.4 | 1.3 | 1.0 | **1.0** |
| | ≥±33% difference from provincial ratio | -32.5 | -11.9 | -22.7 | -23.2 | -22.0 | **-34.3** | -18.8 | 10.7 | 1.2 | **35.9** | 23.7 | -2.7 | |
| Exclusive breastfeeding | Ratio of Year 4 to mean of Year 1–3 | 1.8 | 1.0 | 1.1 | 1.4 | 1.5 | 0.8 | 0.8 | 1.2 | 1.1 | 1.7 | 1.9 | 1.0 | **1.2** |
| | ≥±33% difference from provincial ratio | **47.0** | -19.5 | -7.6 | 19.0 | 24.0 | -30.0 | **-35.9** | -1.0 | -9.5 | **45.4** | **58.9** | -16.8 | |

Data items with ±33%difference between district and provincial ratio are bold

Data source: HMIS/DHIS2

**Table 4. Summary result of number of outliers in selected MNCH data items.**

| Indicators | Rukum West | Rolpa | Pyuthan | Gulmi | Arghakhanchi | Palpa | Nawalparasi West | Rupandehi | Kapilbastu | Dang | Banke | Bardiya |
|---|---|---|---|---|---|---|---|---|---|---|---|---|
| Four ANC visit as per protocol | | 1 | | | 1 | 1 | 1 | | | 1 | | 1 |
| Institutional delivery | 1 | 1 | 1 | | | | 1 | | 1 | | | |
| Women received delivery incentive on transportation | 1 | 1 | 1 | | | 1 | | | | 1 | | |
| Three PNC visit as per protocol | 1 | 1 | 1 | | | 1 | 1 | 1 | | 1 | | 1 |
| Children immunized with MR2 | 2 | 1 | | | | | | | | 1 | | 1 |
| Diraahoea cases treated with ORS and Zinc | 1 | | | 1 | | | 1 | | 1 | | | 1 |
| New Growth Monitoring visit (0–11 Months children) | | 1 | 1 | | 1 | | 1 | | 1 | 1 | 1 | |
| Exclusive Breastfeeding practice | | | | | 1 | | 1 | | 1 | | | |
| Total outliers | 6 | 6 | 4 | 1 | 3 | 3 | 6 | 1 | 4 | 5 | 1 | 4 |
| Average % of moderate outliers (± 2–3 SD from district mean)* | **6.3** | **6.3** | 4.2 | 1.0 | 3.1 | 3.1 | **6.3** | 1.0 | 4.2 | **5.2** | 1.0 | 4.2 |

Districts with ≥ 5% of values that are moderate outliers are bold

*This represents the average percentage of values classified as moderate outliers among the eight data items (calculated as the total number of outliers divided by the total expected reported values, expressed as a percentage

Data source: HMIS/DHIS2

## Consistency between related data

Data items which have a predictable relationship are examined to determine whether the expected relationship exists between those data items. Out of six-pairs of data items compared, three pairs (DPT-HepB-Hib-1st Vs Pneumococcal conjugate vaccine - 1st, institutional delivery Vs delivery presentations, pneumonia cases Vs pneumonia cases treated with antibiotics) that should be equal are within ±10% of each other in all 12 districts. Related data items from the districts that should have equal values did not meet the WHO guidance are: first ANC visit (any time) of women did not equal the pregnant women receiving deworming tablets in women in Rupandehi and Dang districts, four ANC visit as per protocol did not equal the pregnant women receiving 180 iron tablets in Rupandehi district and total diarrhoea cases did not equal with diarrhoea cases treated with ORS and Zinc. Overall, the percent difference for these indicator pairs ranged from 11.4% to 14.1%. (Table 5).

## External consistency of coverage rates

This dimension assessed the level of agreement between two sources of data measuring the same health indicator based on five tracer indicators. Table 6 depicts a comparison of indicator yielded from DHIS2 to the estimates from 2021 DHS survey at the province level. The comparison of all five MNCH indicators falls within the confidence limit or within ±33% of the DHS survey result, indicating a consistent pattern between household survey data and facility-based routine data (Table 6).

## Discussion

Health facility reported routine data are critical for program monitoring, optimizing performance and for planning purposes [27]. We assessed the quality of routine health information for monitoring key Maternal, Newborn and Child Health Indicators of the districts in Lumbini

**Table 5. Consistency between related data in DHIS2.**

| Data elements/district | Rukum East | Rolpa | Pyuthan | Gulmi | Arghakhanchi | Palpa | Nawalparasi West | Rupandehi | Kapilbastu | Dang | Banke | Bardiya |
|---|---|---|---|---|---|---|---|---|---|---|---|---|
| Diphtheria, Tetanus, Pertussis, Hepatitis B and Haemophilus Influenza vaccine (DPT-HepB-Hib)- 1st | 1143 | 4464 | 4866 | 4372 | 3212 | 3783 | 6067 | 19313 | 14435 | 11431 | 12275 | 7490 |
| Pneumococcal conjugate vaccine (PCV)-1st | 1143 | 4448 | 4866 | 4374 | 3213 | 3783 | 6065 | 19313 | 14427 | 11445 | 12251 | 7490 |
| % Difference | 0.00 | 0.36 | 0.00 | -0.05 | -0.03 | 0.00 | 0.03 | 0.00 | 0.06 | -0.12 | 0.20 | 0.00 |
| First ANC (any time) | 1055 | 4734 | 4412 | 3928 | 2813 | 4517 | 7372 | 34846 | 16544 | 12456 | 14669 | 7876 |
| Pregnant women receiving Deworming | 980 | 4327 | 4412 | 3822 | 2833 | 4655 | 7004 | 30529 | 15275 | 11221 | 14316 | 7876 |
| % Difference | 7.7 | 9.4 | 0.0 | 2.8 | -0.7 | -3.0 | 5.3 | **14.1** | 8.3 | **11.0** | 2.5 | 0.0 |
| Four ANC as per protocol | 680 | 3257 | 3084 | 3276 | 2166 | 4143 | 5241 | 22185 | 9290 | 7922 | 9763 | 5597 |
| Pregnant women receiving-180 Iron tablets | 654 | 3269 | 3084 | 3279 | 2187 | 4162 | 5129 | 19785 | 9015 | 7620 | 9023 | 5597 |
| % Difference | 4.0 | -0.4 | 0.0 | -0.1 | -1.0 | -0.5 | 2.2 | **12.1** | 3.1 | 4.0 | 8.2 | 0.0 |
| Institutional Deliveries | 715 | 3304 | 3870 | 2546 | 1436 | 4526 | 3245 | 28858 | 8942 | 8029 | 20390 | 4664 |
| Total delivery presentations | 720 | 3304 | 3870 | 2542 | 1437 | 4527 | 3246 | 28863 | 8956 | 8046 | 20392 | 4664 |
| % Difference | -0.69 | 0.00 | 0.00 | 0.16 | -0.07 | -0.02 | -0.03 | -0.02 | -0.16 | -0.21 | -0.01 | 0.00 |
| Total Pneumonia cases | 972 | 3130 | 1927 | 611 | 526 | 1022 | 354 | 886 | 427 | 1631 | 1474 | 979 |
| Pneumonia treated with antibiotics | 1051 | 3130 | 1927 | 620 | 528 | 972 | 357 | 888 | 434 | 1632 | 1474 | 979 |
| % Difference | -7.52 | 0.00 | 0.00 | -1.45 | -0.38 | 5.14 | -0.84 | -0.23 | -1.61 | -0.06 | 0.00 | 0.00 |
| Total diarrhoea cases | 1842 | 4961 | 3753 | 2644 | 1438 | 2155 | 2598 | 5935 | 8558 | 4250 | 5675 | 3866 |
| Diarrhoea cases treated with Oral Rehydration Solution & Zinc | 1654 | 4961 | 3759 | 2687 | 1490 | 2155 | 2568 | 6119 | 8744 | 4344 | 5649 | 3867 |
| % Difference | **11.4** | 0.0 | -0.2 | -1.6 | -3.5 | 0.0 | 1.2 | -3.0 | -2.1 | -2.2 | 0.5 | 0.0 |

Indicator pairs ≥±10% of each other are bold

Data source: HMIS/DHIS2

Province. We included four MNCH datasets and 23 tracer indicators reflecting maternal and child health services for assessing data quality. Like other studies reviewing routine data, the MNCH data in DHIS2 for the districts in Lumbini Province indicate areas for improvement to fully meet all the defined criteria for internal consistency [28–31], despite meeting the required criteria for timeliness and completeness.

**Table 6. Consistency of data between HMIS/DHIS2 and DHS 2021 survey.**

| Indicators | Health facility coverage rate in 2021/22 | DHS 2022 coverage rate | Ratio of facility to survey rates | ≥33% difference between routine data and survey data |
|---|---|---|---|---|
| Four or more ANC visits | 79.5 | 86.9 | 0.9 | -7.4 |
| Health facility delivery | 94.2 | 84.4 | 1.1 | 9.8 |
| Births assisted by skilled provider | 89.8 | 86.9 | 1.0 | 2.9 |
| BCG coverage | 103.1 | 96.6 | 1.1 | 6.5 |
| Measles Rubella1 | 96.7 | 92.5 | 1.0 | 4.2 |

All the districts achieved the reporting completeness of 98% or more of expected monthly reports for each MNCH datasets during the reference year 2021/22. Although few health facilities in 7 of 12 districts have issue of dataset completeness, no significant gap was observed in achieving the provincial target of 100% completeness of facility completeness in each of the datasets. Observed rate of completeness of MNCH dataset is good comparing with other provinces of Nepal and recent studies conducted in other settings. In India, average completeness levels for selected MNCH indicators were found to be 88.5% [10]. Data completeness was 76% in 17 districts of Ethiopia [32], 86.9% in Kenya [33] and 96.6% in Rwanda [34]. Challenges in achieving completeness of DHIS2 reporting are also noted in other studies as well [32, 35]. Timely submission of reports is crucial as this could have implications for guiding future plans in improving the health of maternal, newborn and child health. Based on this assessment, the provincial average of the timeliness of facility reporting ranged from 94% to 96% across MNCH datasets. Our assessment result of timeliness is observed higher than other provinces of the Nepal [36] as well as the rate reported elsewhere; 78.7% in Kenya [33], 70% in Ethiopia [37] and 46% in Uasin Gishu County Referral Hospital of Kenya, but findings are consistent to studies from Rwanda and Harari region, Ethiopia, where 93.8% and 93.7% timeliness was reported respectively [34]. Comparative higher completeness and timeliness of facility reporting rates of districts in Lumbini province could be attributable to a introduction of vigilant process of DHIS2 data sets assignment by Health Directorate, rigorous data quality review and follow-up through data management committees, six levels of reviews (health facility, LLGs, district, Provincial and federal), shifting of reporting role from municipal health sections to health facilities and knowledge transfer activities, including training and onsite coaching and mentoring on HMIS/DHIS2 [38].

DHIS2 reported data for all 9 MNCH data items in the reference year are consistent over time in 4 of 12 districts as all the selected data elements are within in ±33% difference from the provincial ratio. The introduction of new data quality review interventions, including the establishment of data management committees, the rollout of DHIS2 at the facility level, and relevant knowledge transfer and data monitoring activities in the reference year might also have contributed to inconsistencies in reporting between the reference year and the previous three fiscal years combined. Although the studies assessing consistency over time are limited, few studies in other countries also reported some data quality issue while analyzing trend over the years [39, 40]. Differences in values are obvious over period of time; however, if the differences are so large, it usually suggests data quality issue for further scrutiny. Nevertheless, there is also possibility of an introduction of new programmatic intervention which might have contributed to significant increase in values from one year to the next. To illustrate, the phase-wise rollout of the postnatal care home visit program by the government contributed [41] to a gradual increment in PNC coverage from FY 2018/19 to 2021/22. Therefore, there might be programmatic implications in not maintaining consistency over time for the specific indicator.

The outlier analysis provided valuable insights in addition to consistency over time reflected above. The assessment included 8 MNCH indicators for outlier analysis in which average percentage of moderate outliers ranged from 1–6% in a reference year, with four districts reported ≥5% monthly values that were moderate outliers for the selected MNCH indicators. Significant variations in these districts need further assessment at the facility level to confirm that these variations are legitimate or there is a serious data quality issue. Nevertheless, none of the data elements are prone to extreme outliers in any districts. These findings are better while comparing the result with other similar studies assessing outliers [42]. In contrast, no districts had reported ≥5% monthly values with moderate or extreme outliers in a study conducted in Ruwanda [43] and Ghana [40].

Internal consistency between six-pairs of data items that are expected to show similar patterns of behaviour are compared and found that three pairs are within ±10% of each other in

all 12 districts. Nevertheless, two districts did not meet the criteria in: first ANC visit (any time) of women Vs pregnant women receiving deworming tablets, four ANC visit as per protocol Vs women receiving 180 iron and total diarrhoea cases Vs treated with ORS and Zinc. Findings explored the room for improvement to ensure internal consistency, preferably through sustaining integrated data review and feedback mechanisms at multiple levels. Related indicators that should show expected numerical relationship did not meet the WHO guidance are also found in other studies as well. A study conducted in Nigeria also showed that no one of the priority MNCH indicators compared showed the anticipated numerical relationship across all facilities [39].

We analyzed the external consistency of tracer MNCH indicators between DHIS2 and 2021 survey estimates at the province level. Comparison between the coverage rates of tracer indicators fall within ±33% of the DHS survey result and none of the coverage rates are flagged. In contrast to this result, greater degree of discordance was found between DHIS2 based facility reports and household survey data in a studies conducted in other settings [35, 44].

This assessment has certain limitations. The paper-based HMIS reports were not verified with DHIS2 data, as in other studies [45–47]. Advanced analysis techniques, such as t-tests or ANOVA, were not utilized in the review to evaluate temporal changes in the related indicators to make the manuscript understandable up to the health facility level. In assessing external consistency, it is important to note that the DHS survey and facility-based DHIS2 data used in this assessment should not be considered the gold standard; rather, they were compared to provide relevant references for assessing the external consistency of routine data in DHIS2. It is imperative for health managers to identify consistency between multiple sources of data in order to make informed decisions about their appropriate use. Further, as the survey coverage rates for the selected tracer indicators are not available at the district level, we have only assessed the external consistency of data at province level.

This assessment extended the evidence that the health facility data available in DHIS2 is complete and reported on time considering national benchmark. Data is credible for use, although there is room for improvement in maintaining internal consistency of reported data. Findings of this assessment can be useful to researchers for standardizing published evidence relating to MNCH related routine data quality and providing evidence to data managers to develop, focus and evaluate facility-based data quality initiatives and ultimately contributing the MNCH outcomes.

The need for quality data is crucial, specifically for populations with greater risk of mortality and morbidity, such as pregnant and lactating women, newborn, and children. Therefore, health system should design multiple strategies and be watchful to maintain complete, timely, accurate and consistent data. Routine data review, feedback, and supervision at all levels of the health system have been proven essential to optimize routine data for monitoring [28, 29, 31, 48]. Data management committees formed at the various levels should be strengthened for routine data review, practice of knowledge transfer activities and information use at local level (i.e., where data is collected) should be promoted, system- inbuilt data validation mechanism of DHIS2 should be strengthened and data quality assessment systems should greatly be encouraged. As outlier analysis revealed significant variations in data for some districts, with ≥5% occurrence of moderate outliers for the selected data items, further assessment is warranted to confirm whether these variations are legitimate or indicative of issues in the quality of data.

## Ethical consideration

This assessment obtained permission to access the DHIS2 platform and consent for the publication of the manuscript from the Provincial Health Directorate Office of the Lumbini

Province government (Ref No. 2759). This analysis considered non-human subject assessment as only aggregated secondary source of data which can be accessible on request or available in the DHIS2 domain were included in the assessment. No Identifiers such as name of the individuals were considered for the assessment.

## Supporting information

**S1 Table. Outlier analysis of maternal health indicators.**
(DOCX)

**S2 Table. Outlier analysis of child health indicators.**
(DOCX)

**S1 File. Data analysis results.**
(XLSX)

## Acknowledgments

The assessment team is thankful to the IHIMS Section- DOHS and Health Directorate for their support in retrieval of data from DHIS2. We are also thankful to Madhav Chaulagain, Lalita Timalsina and Nilakantha Gautam for their review and feedback in finalizing the manuscript.

## Author Contributions

**Conceptualization:** Keshab Sanjel.

**Formal analysis:** Keshab Sanjel, Swadesh Gurung.

**Software:** Man Bahadur Oli.

**Writing – original draft:** Keshab Sanjel, Man Bahadur Oli.

**Writing – review & editing:** Keshab Sanjel, Shiv Lal Sharma, Swadesh Gurung, Samikshya Singh, Tuk Prasad Pokhrel.

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
