## [Decision Letter · Decision Letter 0]

12 Apr 2023

PONE-D-23-07291Quality of Routine Health Information for Monitoring Maternal, Newborn and Child Health Indicators: An Analysis of DHIS2 Data in Lumbini Province, NepalPLOS ONE

Dear Dr. Sanjel,

Thank you for submitting your manuscript to PLOS ONE. After careful consideration, we feel that it has merit but does not fully meet PLOS ONE’s publication criteria as it currently stands. Therefore, we invite you to submit a revised version of the manuscript that addresses the points raised during the review process.

We look forward to receiving your revised manuscript.

Kind regards,

Kanchan Thapa, MPH, MPhil

Academic Editor

PLOS ONE

Additional Editor comments: 

I enjoyed reading your paper. The paper adds news value in the field of Public Health in Nepal and also provides important information for global readers. The paper went through extensive reviews and has several comments from the reviewer, I request you improve your paper as per the reviewer's comments.

Furthermore, the paper relies on routine data information from a routine data source of the Ministry of Health. As the data is collected through the routine health information management system of the government of Nepal. I request you provide a letter from the government (Ministry of Health) stating that they have provided consent for the publication of their information.

At this stage, I echo all the reviewer comments for further action. Please take care of all four reviewers’ comments. 

Note: HTML markup is below. Please do not edit.]

Reviewers' comments:

Reviewer's Responses to Questions

**Comments to the Author**

1. Is the manuscript technically sound, and do the data support the conclusions?

Reviewer #1: No

Reviewer #2: Partly

Reviewer #3: Yes

Reviewer #4: Partly

2. Has the statistical analysis been performed appropriately and rigorously? 

Reviewer #1: No

Reviewer #2: Yes

Reviewer #3: No

Reviewer #4: Yes

3. Have the authors made all data underlying the findings in their manuscript fully available?

Reviewer #1: Yes

Reviewer #2: No

Reviewer #3: Yes

Reviewer #4: Yes

4. Is the manuscript presented in an intelligible fashion and written in standard English?

Reviewer #1: Yes

Reviewer #2: Yes

Reviewer #3: Yes

Reviewer #4: Yes

5. Review Comments to the Author

Reviewer #1: The manuscript entitled “Quality of Routine Health Information for Monitoring Maternal, Newborn and Child Health Indicators: An Analysis of DHIS2 Data in Lumbini Province, Nepal” was interesting. The following comments can help the authors to improve it:

1- In the abstract, please use the full form of MNCH.

2- In the abstract, results and conclusion, it is not what MNCH datasets and indicators were and how they were compared. What were the data items? Which data were not consistent?

3- Please choose appropriate keywords based on the MeSH terms.

4- In the introduction section, there is no adequate information about data quality and related indicators. Moreover, studies related to evaluating data quality of health information systems need to be reviewed and added to the introduction section.

5- In the methods section, please provide a reference for lines 126-128.

6- The authors need to explain more about Table 1. For me, it is not still clear how the data elements and indicators have been selected.

7- In Table 2, did the authors look for data elements completeness or just the completeness of reporting on a monthly basis? This question is about timeliness and consistency, too.

8- About the consistency of the mean values, the authors need to explain why it was important for them.

9- What do the authors mean by the Indicator-pairs in and six-pairs of data items Table 2?

10- Please remove unnecessary tables.

11- I think in the current study, the quality of health information has not been evaluated. If the authors did that, more information about data elements and quality indicators should be provided in the manuscript. Instead they assessed quality of facility reporting. Therefore, I suggest the authors to revise the title and the content of the manuscript to focus on quality of facility reporting not health data or information quality.

12- Apart from the statistical analysis, what are the innovative side of the research?

13- Please re-check the referencing style.

Reviewer #2: The authors sought to assess the Quality of Routine Health Information for Monitoring Maternal, Newborn and

Child Health Indicators: An Analysis of DHIS2 Data in Lumbini Province, Nepal.

Authors did not thoroughly explain the research design deployed for their study. It is important for the scientific community to appreciate the research design deployed in order to draw appropriate links to the content of the study.

The dimensions of data quality that the authors focused on were only completeness of the data, timeliness of the data and lastly the consistency/reliability of the data. However, using the same WHO data quality toolkit Module1 that the authors relied on, important dimension of data quality was conspicuously missing in the whole study. This dimension is the accuracy/validity of the data which is very important in unraveling whether the data faithfully reflects the actual level of service delivery conducted in the various health facilities included in the study. The manuscript should be modified to incorporate this important element in the assessment of quality data.

The maternal indicators highlighted by the authors did not include HIV (with its attendant number of HIV positive mothers receiving ART), TB (with TB positive mothers receiving treatment) as well as mothers who tested positive for malaria either by microscopy or RDT) as these are very important maternal health indicators highlighted by the WHO data quality toolkit Module1. Once authors sourced their secondary data from the DHIS2, it is easier to retrieve the aforementioned variables.

Authors compared six pair of data in order to establish their relationship or association. The statistical basis of the comparison is defective as authors after their initial analysis as captured in table 9 ought to have deployed regression or other correlative statistical models to consolidate the association or otherwise observed.

Authors must correct the spelling of Ethical as captured in their Ethical Considerations.

Authors must modify their study to capture the comments highlighted above.

Reviewer #3: The paper is well written with interesting topic.

The last sample size and national coverage is outstanding. However, some comments might improve this manuscript.

The comment are as follows:

1- To reduce number of tables as it provides a distraction.

2- Add more advance analysis such as t test or Anova to assess the temporal change in the related indicators to check how significant is the difference in the KPI.

3- Add more reference to the discussion part.

4- Were this study follows any guidelines such as STROBE? If no try to adapt it to assure the study is conducted in acceptable scientific writing.

Reviewer #4: The specific aim of the study was not clear. The methodology used to assessed the quality of reports did not meet the WHO Data Quality Review (DQR) Standards. Data quality assessment by WHO methodology is in two major phases

1. Desk Review

2.Site Assessment

By this article, it was only phase 1 that was carried out. I therefore suggest the author should at least sample the facilities for site assessment to authenticate the reports entered into DHIMS2 since the WHO data quality toolkit is the refeenced tool used in the study

6. PLOS authors have the option to publish the peer review history of their article (what does this mean?). If published, this will include your full peer review and any attached files.

Reviewer #1: No

Reviewer #2: **Yes: **ANTWI JOSEPH BARIMAH

Reviewer #3: No

Reviewer #4: No

---

## [Author Response · Author response to Decision Letter 0]

11 Jul 2023

Authors’ responses to review comments:

First of all, the authors are grateful to the editors and reviewers for taking their time and providing these interesting and constructive comments that help us improve the quality of our manuscript. With this, we have tried to address all the comments point-by-point in the revised manuscript as follows.

Responses to comments by Reviewer #1: 

1- In the abstract, please use the full form of MNCH.

Response: By accepting the comment, we mentioned the full form of MNCH in the revised manuscript. 

2- In the abstract, results and conclusion, it is not what MNCH datasets and indicators were and how they were compared. What were the data items? Which data were not consistent?

Response: Thank you for this comment. We assessed the completeness and timeliness of facility reporting (district wise) considering datasets as listed in DHIS2 (listed datasets relating to MNCH: Safe motherhood, Immunization, Nutrition and CBIMNCI). For the consistency, we selected key MNCH data items (as listed in Table 1). Based on the comment, we slightly modified the manuscript.

3- Please choose appropriate keywords based on the MeSH terms.

Response: Thank you for the suggestion. We revised the keywords and used appropriate keywords based on MeSH terms. 

4- In the introduction section, there is no adequate information about data quality and related indicators. Moreover, studies related to evaluating data quality of health information systems need to be reviewed and added to the introduction section.

Response: This comment is well taken. Information about data quality and studies related to data quality assessments are further reviewed and added to the introduction section. 

5- In the methods section, please provide a reference for lines 126-128.

Response: This is well taken, and references added for the lines 126-128.

6- The authors need to explain more about Table 1. For me, it is not still clear how the data elements and indicators have been selected.

Response: Thank you for this comment. Logic for selecting the data items listed in Table 1 are now clarified in the revised manuscript. Authors now mentioned: The selection of data items takes into account the MNCH continuum of care approach, their ability to cross-verify and measure data quality issues, as well as their importance in program monitoring and evaluation, as stated in national and provincial policy documents.

7- In Table 2, did the authors look for data elements completeness or just the completeness of reporting on a monthly basis? This question is about timeliness and consistency, too.

Response: We thank the reviewer for this valid comment. In Table 2, the authors assessed the timeliness and completeness of facility reporting of the MNCH-related datasets on a monthly basis. For consistency, the authors enrolled the data items (Table 1) and assessed the consistency over time, outliers in the current year, consistency between related data and external consistency of data. The Authors explicitly explained the measurement approaches in the revised manuscript. 

8- About the consistency of the mean values, the authors need to explain why it was important for them.

Response: The importance of assessing consistency of the mean values is now well explained in the revised manuscript. The authors explicitly explained: the consistency of the time indicator was assessed to observe whether the differences in values are expected from one year to the next. In the case where there is an existence of larger difference, it suggests the need for further scrutiny. While large differences usually suggest some type of reporting error, it is also possible that the introduction of a new intervention might have contributed to a significant percentage increase in indicator values from one year to the next.

9- What do the authors mean by the indicator-pairs in and six-pairs of data items Table 2?

Response: Thank you for this valid question. Indicator-pairs in the Table 2 refer to a set of related data items or attributes that are expected to be consistent and maintain predictable relationship with each other. When assessing the quality of data, Authors examined indicator-pairs to identify any discrepancies that may exist within the two data items. By comparing the values of related data items, we can detect potential errors and inconsistencies. To analyze the consistency between related data, we enrolled six-pairs of data items and summarized if they exhibit the predicted relationship. 

10- Please remove unnecessary tables.

Response: Thank you again for this valid suggestion. Unnecessary tables (Table 6 and 7 in the submitted manuscript) are transferred to supporting information section (named as S1 Table and S2 Table). 

11- I think in the current study, the quality of health information has not been evaluated. If the authors did that, more information about data elements and quality indicators should be provided in the manuscript. Instead they assessed quality of facility reporting. Therefore, I suggest the authors to revise the title and the content of the manuscript to focus on quality of facility reporting not health data or information quality.

12- Apart from the statistical analysis, what are the innovative side of the research?

Response: Thank you for your suggestion. The authors made slight revisions to both the title and content of the manuscript, clearly stating that the review focuses on the quality of facility reporting. In the context of Nepal, there is limited evidence indicating that data quality assurance processes have been implemented for health facility data. At the sub-national level, specifically in Lumbini Province, multiple interventions for data quality assurance were carried out by both provincial and local authorities. However, the level of data quality remained unknown to the data generation and supervising authorities. Therefore, this assessment provides evidence to sub-national governments regarding the data quality status of health facilities. Additionally, this study aims to promote the practice of routine data quality reviews at the data generation level.

13- Please re-check the referencing style.

Response: Thank you for this valid comment. We re-checked and managed the referencing style as per the guideline. 

Responses to comments by Reviewer #2: 

1- Authors did not thoroughly explain the research design deployed for their study. It is important for the scientific community to appreciate the research design deployed in order to draw appropriate links to the content of the study.

Response: Thank you for the suggestion. Authors now mentioned the research design for the study. In the methods section, the Authors also elaborated the basis for indicator selection and analysis processes. 

2- The dimensions of data quality that the authors focused on were only completeness of the data, timeliness of the data and lastly the consistency/reliability of the data. However, using the same WHO data quality toolkit Module1 that the authors relied on, important dimension of data quality was conspicuously missing in the whole study. This dimension is the accuracy/validity of the data which is very important in unraveling whether the data faithfully reflects the actual level of service delivery conducted in the various health facilities included in the study. The manuscript should be modified to incorporate this important element in the assessment of quality data.

Response: We fully agree with the fact that we solely focused on the completeness and timeliness of MNCH datasets, as well as the consistency and reliability of key MNCH data items. We clearly mentioned in the study that it was based on a desk review of DHIS2 data. Due to the limitations of time and resources, it appears to be impossible to measure the accuracy of the data since it requires visits to health facilities and verification of data in multiple stages. This limitation was therefore mentioned in the discussion section of the manuscript. 

3- The maternal indicators highlighted by the authors did not include HIV (with its attendant number of HIV positive mothers receiving ART), TB (with TB positive mothers receiving treatment) as well as mothers who tested positive for malaria either by microscopy or RDT) as these are very important maternal health indicators highlighted by the WHO data quality toolkit Module1. Once authors sourced their secondary data from the DHIS2, it is easier to retrieve the aforementioned variables.

Response: While we did refer to the WHO DQA module for selecting DQA indicators and assessing the consistency of reported data, this review was not entirely based on the WHO Data Quality Review (DQR) Standards. For example, we were unable to assess the timeliness and completeness of specific data items because DHIS2 does not have a function to measure this. Instead, we evaluated the completeness and timeliness of dataset-specific information as reflected in the DHIS2 platform. We selected the indicators take into account the MNCH continuum of care approach, their ability to cross-verify and measure data quality issues (as specified in data verification guide and DHIS2 data quality app), as well as their importance in program monitoring and evaluation, as stated in national and provincial policy documents. 

4- Authors compared six pair of data in order to establish their relationship or association. The statistical basis of the comparison is defective as authors after their initial analysis as captured in table 9 ought to have deployed regression or other correlative statistical models to consolidate the association or otherwise observed.

Response: We analyzed six pairs of data to measure the consistency between related data. In order to review this, we defined the metrics according to the WHO's module on discrete desk review of data quality, and these metrics align with the data quality review practices utilized at the province and local levels in Lumbini province. Advanced statistical analysis was not employed in the assessment to ensure the manuscript remains understandable at the data generation level. The inability to utilize advanced analysis for this review is mentioned in the limitation section of the manuscript as well.

5- Authors must correct the spelling of Ethical as captured in their Ethical Considerations.

Response: Thank you. We corrected the spelling. 

Responses to comments by Reviewer #3: 

1- To reduce number of tables as it provides a distraction.

Response: Thank you, we reduced the number of tables as suggested. 

2- Add more advance analysis such as t test or Anova to assess the temporal change in the related indicators to check how significant is the difference in the KPI.

Response: Thank you for your suggestion. We analyzed the three data quality dimensions (timeliness of facility reporting, consistency of facility reporting and the consistency of data). We relied on DHIS2 Pivot Table for the timeliness and completeness analysis, as this is the way that all the level of health system relies on defining HMIS reporting status. To review the consistency of data, we defined the metrics as per the WHO’s Module on Discrete desk review of data quality and these metrics are consistent with the data quality review practices employed by province and local levels in Lumbini province. More advance statistical analysis was not employed in the assessment to make the manuscript understandable up to health facility level. The unavailability to employ advanced analysis for this review are therefore mentioned in the limitation section of the manuscript as well. 

3- Add more reference to the discussion part.

Response: Thank you for the feedback. We tried to review more studies and included references in the discussion section. Nevertheless, due to the limited availability of studies on routine MNCH data quality review, the authors may not be fully able to add more references to the pre-existing literature to demonstrate how the findings either align with or differ from previous research in some of the findings.

4- Were this study follows any guidelines such as STROBE? If no try to adapt it to assure the study is conducted in acceptable scientific writing.

Response: We followed the submission guideline developed by PLOS One journal. Referring to the Strengthening the Reporting of Observational studies in Epidemiology (STROBE) guideline also, we assure that the review is conducted in acceptable scientific writing.

Responses to comments by Reviewer #4: 

1- The specific aim of the study was not clear. The methodology used to assessed the quality of reports did not meet the WHO Data Quality Review (DQR) Standards. Data quality assessment by WHO methodology is in two major phases

1. Desk Review

2.Site Assessment

By this article, it was only phase 1 that was carried out. I therefore suggest the author should at least sample the facilities for site assessment to authenticate the reports entered into DHIMS2 since the WHO data quality toolkit is the referenced tool used in the study

Response: We specified the aim of the study as a review of facility reported MNCH data. We conducted a desk review for this purpose and explicitly stated in the limitations section that we did not verify the paper based HMIS reports with DHIS2 data. While we did refer to the WHO DQA module for selecting DQA indicators and assessing the consistency of reported data, this review was not entirely based on the WHO Data Quality Review (DQR) Standards. For example, we were unable to assess the timeliness and completeness of specific data items because DHIS2 does not have a function to measure this. Instead, we evaluated the completeness and timeliness of dataset-specific information as reflected in the DHIS2 platform. Additionally, due to time and resource constraints, we did not conduct site visits, and these limitations are clearly discussed in the corresponding section.

Additional Editor comments: 

Furthermore, the paper relies on routine data information from a routine data source of the Ministry of Health. As the data is collected through the routine health information management system of the government of Nepal. I request you provide a letter from the government (Ministry of Health) stating that they have provided consent for the publication of their information.

Response: As already mentioned in the manuscript, we received consent to access DHIS2 platform and publish the data from provincial Health Directorate under Ministry of Health, Lumbini Province. The Health Directorate in Lumbini Province is responsible for the overall information management in the province including DHIS2 user management. The authors also attached a letter from the government mentioning that they have provided consent for the publication of DHIS2 stored data. 

Furthermore, changes have been made to the order of authors and are reflected in the Authorship Change Form. Please find the form attached under the "Others" category.

Thank you so much! 

The Authors

---

## [Decision Letter · Decision Letter 1]

12 Sep 2023

PONE-D-23-07291R1Quality of Routine Health Facility Data for Monitoring Maternal, Newborn and Child Health Indicators: A Desk Review of DHIS2 Data in Lumbini Province, NepalPLOS ONE

Dear Dr. Sanjel,

Thank you for submitting your manuscript to PLOS ONE. After careful consideration, we feel that it has merit but does not fully meet PLOS ONE’s publication criteria as it currently stands. Therefore, we invite you to submit a revised version of the manuscript that addresses the points raised during the review process. Thank you for improving your paper significantly. You are suggested to work further on your paper based on comments of reviewer. Please address the comments from the reviewers which they have provided in .doc file. 

We look forward to receiving your revised manuscript.

Kind regards,

Kanchan Thapa, MPH, MPhil

Academic Editor

PLOS ONE

Additional Editor Comments:

Thank you for improving your paper significantly. You are suggested to work further on your paper based on comments of reviewer.

Reviewers' comments:

Reviewer's Responses to Questions

**Comments to the Author**

1. If the authors have adequately addressed your comments raised in a previous round of review and you feel that this manuscript is now acceptable for publication, you may indicate that here to bypass the “Comments to the Author” section, enter your conflict of interest statement in the “Confidential to Editor” section, and submit your "Accept" recommendation.

Reviewer #5: (No Response)

Reviewer #6: All comments have been addressed

Reviewer #7: All comments have been addressed

Reviewer #8: (No Response)

Reviewer #9: (No Response)

2. Is the manuscript technically sound, and do the data support the conclusions?

Reviewer #5: No

Reviewer #6: (No Response)

Reviewer #7: Yes

Reviewer #8: Yes

Reviewer #9: Partly

3. Has the statistical analysis been performed appropriately and rigorously? 

Reviewer #5: No

Reviewer #6: I Don't Know

Reviewer #7: I Don't Know

Reviewer #8: No

Reviewer #9: No

4. Have the authors made all data underlying the findings in their manuscript fully available?

Reviewer #5: Yes

Reviewer #6: Yes

Reviewer #7: Yes

Reviewer #8: Yes

Reviewer #9: Yes

5. Is the manuscript presented in an intelligible fashion and written in standard English?

Reviewer #5: Yes

Reviewer #6: No

Reviewer #7: Yes

Reviewer #8: Yes

Reviewer #9: Yes

6. Review Comments to the Author

Reviewer #5: Although the authors have made considerable efforts to ensure the precision and rigor of this manuscript, I find myself not entirely satisfied with the methodology and analysis sections. Specifically, I would like to see more robust statistical analyses in the results section. For instance, incorporating powerful statistical tools like Bland-Altman Analysis for continuous numerical measurements in both datasets and employing the Kappa Statistic to measure agreement between the two systems (paper-based and digital) could strengthen the findings significantly. I kindly request the authors to enhance the analysis using these powerful statistical methods to achieve more robust and compelling results.

Reviewer #6: MORE DESCRIPTION OF THE MANUFACTURING PROGRAM WILL ADD INFORMATION TO THE READER FOR PEOPLE IN OTHER COUNTRIES

Reviewer #7: the author has adjusted the manuscript according to the suggestions of the previous reviewer's input and corrected the manuscript. well done

Reviewer #8: (No Response)

Reviewer #9: The study conducted is a quality assessment of maternal and child health indicators in Lumbini Province, Nepal. Factors considered were timeliness, completeness, and consistency of the data available, accessed through the District Health Information Software (DHIS2).

Reviewer Comments:

1. Abstract:

a) Methods section: My suggestion is to not label the study as cross-sectional. The study is ecological in nature since monthly aggregated data is utilized and there is no access to individual data. Another option is to use the term “descriptive study”.

b) Methods section: In a sentence, please mention the number of data elements selected for each objective. Readers might first assume that all 23 data items were used for all objectives, which isn’t the case.

2. Main Text:

a) Methods: Same comment as 1a regarding study design.

b) Line 138: Please add a comma before “and Safe Motherhood”. Otherwise, “Nutrition and Safe Motherhood” is interpreted as a single dataset.

c) For all tables, please add footnotes explaining the abbreviations used in them. Readers will find it cumbersome to search for the abbreviations in the main text. For example, in Table 1 please explain ANC, PNC, what is the meaning of ‘x’ in the column etc. Also, mention the relevant years for each table.

d) Table 2: While assessing outliers the authors have selected ±2SD. This method is apt for normally distributed data. No tests of normality or graphical visualizations were conducted and selecting ±2SD as the benchmark could be problematic. It would be better to conduct some basic tests to verify normality which isn’t complex. Additionally, please mention if selecting ±33% and ±10% as the benchmark for other metrics of consistency is specifically mentioned in the WHO data quality report card (or any other reference). If not, wjhat was the reasoning behind the criteria?

e) In the first paragraph of the results section (lines 183-184), it is mentioned that “5 out of 12 districts have 100% completeness for nutrition and IMNCI datasets (in each) and 4 out of 10 districts have 100% completeness for safe motherhood datasets”. However, in Table 3, I count 7 provinces with 100% completion for nutrition and IMNCI individually. Similarly, the results for safe motherhood do not match. Table 3 shows 8 out of 12 districts with 100% completion (and not 4 out of 10). If the total districts for safe motherhood were 10, identify the dropped districts in the table/text and provide a brief explanation as to the reason for exclusion. Please correct/verify the results.

f) While reporting timeliness the authors mention “By district, 9 out of 12 districts have more than 90% reporting rate on time for the selected datasets.” By my calculations, all districts, except Rukum East, have an average above 90%, i.e. 11 out of 12 have more than a 90% score. Please correct/verify.

g) Table 6: It wasn’t clear to me how the results reported in the final row were arrived at after calculating the total outliers in every district. Provide a brief footnote with an explanation.

h) Line 331: The sentence mentions “no one of the coverage rates are flagged” instead of “none”.

7. PLOS authors have the option to publish the peer review history of their article (what does this mean?). If published, this will include your full peer review and any attached files.

Reviewer #5: **Yes: **Laxman Datt Bhatt

Reviewer #6: No

Reviewer #7: No

Reviewer #8: No

Reviewer #9: No

---

## [Author Response · Author response to Decision Letter 1]

29 Nov 2023

Authors’ responses to review comments:

 At first, the authors are grateful to the editors and reviewers for taking their time and providing these interesting and constructive comments that help us improve the quality of our manuscript. With this, we have tried to address all the comments point-by-point in the revised manuscript as follows.

Abstract

Conclusion:

1. Line 48 – Do not lead your readers wondering at this point. What specifically can be done to maintain consistency of data over time? 

Response: Thank you for this question. Specific recommendations to maintain the consistency of data over time was mentioned in the revised manuscript as suggested. 

Main Body

Introduction: 

2. Line 90 - consider inserting ……..reported data might be subjected……

3. Line 96 - I suggest review should be changed to assess.

Response: Updated the manuscript as suggested.

Study setting:

4. Line 108 – why is skilled birth attendant in caps? Consider changing it to lower case.

5. Line 109 – change first letter of incidence to lower case.

Response: Revised as suggested 

Study design and data sources:

6. Line 124 – indicate the previous years you are referring to. This should be applicable to the entire document.

 Response: Indicated previous years: FY 2018/19 to 2020/21 clearly in the revised manuscript. 

Data analysis:

7. Kindly expatiate on how data analysis was carried out.

8. Could authors provide the mean values for the preceding years? 

9. The authors’ response to reviewer 1 comments 8 and 9 should be captured under the current heading (data analysis) for better clarity of how you proceeded with data analysis.

10. In addition to the authors’ explanation to the importance of the consistency of the mean values, you could also allude to the fact that consistency of the mean values is an indicator of reliability-meaning the greater probability that data source is trustworthy.

Response: 

Metrics for data analysis of each domain are clearly mentioned in the Table 2 of manuscript. Analysis process employed are also mentioned in the narrative. 

Mean values of preceding years are not separately mentioned in the manuscript, but the ratio of Year 4 to mean of Year 1-3 ((2021/22 to mean of 2018/19- 2020/21) was mentioned to analyze consistency over time. As per the comment, we would also like to mention the mean values of preceding years hereunder: 

Organisation unit / data items Mean values for the preceding years (2018/19- 2020/21)

 4ANC as per protocol Institutional delivery Received incentive on transportation 4 PNC as per protocol Children immunized with BCG MR2 coverage Diarrhoea cases treated with ORS and Zinc New growth monitoring visit (0-11) Exclusive breastfeeding 

RUKUM EAST 561.67 616.0 581.3 416.0 1192.0 1127.0 1758.0 1610.0 550.0

ROLPA 3172.33 3479.3 3377.3 1973.0 5190.3 4821.0 6116.0 5532.3 3164.0

PYUTHAN 3189.33 3936.7 3936.3 1737.7 4959.3 4677.7 5280.0 6075.7 3592.0

GULMI 3486.67 2568.3 3214.7 1447.0 4269.3 4469.7 3560.3 6286.0 3032.7

ARGHAKHANCHI 2056.33 1377.3 1212.3 490.7 3105.0 3178.3 1638.0 4419.3 2204.0

PALPA 4415.67 5447.0 2700.3 1352.0 4127.0 4023.3 3212.7 4997.3 2719.7

NAWALPARASI WEST 4848.67 3189.0 2989.0 1155.3 5916.0 6364.0 3377.3 7618.7 6522.7

RUPANDEHI 16898.67 27621.3 17260.3 5877.3 26254.0 18448.3 7423.3 21447.0 9683.7

KAPILBASTU 7865.00 7477.3 6593.0 1870.3 13507.3 11614.7 11131.7 12312.3 7457.7

DANG 7666.00 8566.7 9492.0 2778.7 12425.0 11300.3 5509.0 10425.3 4356.0

BANKE 8506.00 19194.3 16671.3 3741.0 13391.3 10253.0 6826.0 9217.7 3719.7

BARDIYA 6181.33 5456.7 5205.0 3193.3 7911.7 7580.0 4385.3 8177.7 4072.0

Lumbini Province 68847.67 88930.0 73233.0 26032.3 102248.3 87857.3 60217.7 98119.3 51074.0

The authors’ response to reviewer 1 comments 8 and 9 are clearly mentioned under the data analysis heading in the revised manuscript.

Authors also included the recommended fact by the reviewer that the consistency of the mean values is an indicator of reliability, meaning a greater probability that the data source is trustworthy.

Results:

Completeness of maternal, newborn and child health datasets &Timeliness of maternal, newborn and child health datasets:

11. Since the report is disaggregated by district, (refer to tables 3 and 4) you should consider reporting the variations based on the districts. These variations could give an indication as to where the error is or source of data limitations within the province. 

12. Provide plausible reasons for the variations observed.

13. Please indicate the sources of data for all your tables at the bottom of each table.

14. Make the province result distinct by separating it with a thick line or write the values in bold. 

Response: 

The authors accepted the comments and revised the manuscript as suggested. In the narrative of reporting completeness and timeliness, the variations between the districts are clearly mentioned. Additionally, plausible reasons for the variations are also included, specifically in the discussion section. The authors clearly indicated the sources of data for all figures and tables. Province results are written in bold to separate the values from district values.

Consistency over time:

15. Move Lines 188 - 192 to the data analysis section.

16. Line 195-196 – the sentence should be preceded or end with at the province level to ensure a distinction between province and district level information.

Response: Addressed the comments

Accuracy of event reporting: outliers in the reference year:

17. Line 210 – Authors should consider inserting values from the table to show your results. This is also applicable to the entire result section. Where necessary, insert some values from the result table as illustration so that your readers can follow you well.

Response: Thank you for this valid comment. Authors now inserted the results from tables in the narrative as relevant.

Discussion:

18. Line 240 – The authors should consider changing reviewed to assessed

Response: Addressed

19. Line 255 – Authors should check the spelling if you are referring to the country in Africa called Rwanda and throughout the entire document. 

Response: Checked and corrected 

20. Line 275 – there is an omission in the sentence, kindly check and revise accordingly.

Response: Checked and corrected 

21. Lines 280-282: At this point of your study, we expect the authors to be emphatic about what is happening in the province. It will be great and interesting to find this out and report it in your study findings.

Response: Thank you for this valid comment. Authors mentioned the programmatic implication of the introduction of the PNC home visit program in not maintaining consistency over time for the data item - PNC visits as per protocol.

22. Line 288 - …..significant variations in these districts need further assessment……..This could be a very good point for your recommendation section.

Response: This point has been added in the recommendation section. 

23. Lines 312-313: Do the authors consider these tests relevant for the study? If yes, why did they not perform the tests and if no why were they not done or why did the authors bring it up as a limitation? It is not appropriate to just indicate that you did not do it, provide justification for it.

Response: Authors do not actually consider these tests compulsory for such a review. The authors added this limitation based on suggestions from the first reviewers. In fact, the authors want this document to remain understandable at the data generation level (i.e., health facility and first-level data managers), where complex statistical tests are less relevant. The reason is now clearly mentioned in the manuscript.

24. Lines 315-317: If this not the gold standard, what is the gold standard and why was is not used in this study. Justify why you had to use the data regardless of the shortfalls (not meeting the gold standard)?

Response: Authors wanted to make it clear that two sources of data were compared to provide relevant references for assessing the external consistency of routine data, even though DHS is a household sample survey and DHIS2 reported data are facility-reported data. For the comparison, there could also be space to compare the DHIS2 data with other facility-based survey data. However, in the case of Nepal, no such surveys were conducted, allowing for the comparison of MNCH-related indicators.

Reviewer #5: Although the authors have made considerable efforts to ensure the precision and rigor of this manuscript, I find myself not entirely satisfied with the methodology and analysis sections. Specifically, I would like to see more robust statistical analyses in the results section. For instance, incorporating powerful statistical tools like Bland-Altman Analysis for continuous numerical measurements in both datasets and employing the Kappa Statistic to measure agreement between the two systems (paper-based and digital) could strengthen the findings significantly. I kindly request the authors to enhance the analysis using these powerful statistical methods to achieve more robust and compelling results.

Response: As suggested by the reviewer, the analysis section has been made more comprehensive by including measurement metrics and additional information in the analysis tables. The authors aimed to simplify the analysis to ensure understanding at the data generation level (i.e., health facility and first-level data managers), where complex statistical tests are less relevant. The study's objective was specified as a review of facility-reported MNCH data. To achieve this, a desk review was conducted. It is explicitly stated in the limitations section that the paper-based HMIS reports were not verified with DHIS2 data.

In line 17-19

DHIS2 is predominantly utilized by low and middle-income countries, making the statement more specific and targeted to its primary user base would be good.

Response: Ay accepting the comment, the authors revised the manuscript. 

Line 67-69

While your sentence effectively highlights specific data quality issues in information systems, I recommend a slight revision for clarity and conciseness. Consider rephrasing it as follows:

Specific data quality issues may arise, including incomplete, inconsistent, and irrelevant data, as well as imprecise estimates of the target population for coverage. These issues could limit the usefulness of the data for decision-makers.

Response: This comment is well taken and revised as suggested. 

Line 69-79

The authors pointed out that discrepancies between paper-based report findings and routine health information system findings. However, they did not provide any background information about the limitations of both systems and the existing gaps between them. I encourage them to explore and elaborate on the existing gaps between these two systems. This will significantly enhance the depth and credibility of their research, enabling readers to better understand the implications of the observed discrepancies.

Response: Authors mentioned about the observed discrepancies between the coverage estimated derived from routine information systems and population based surveys. To further elaborate, Authors discussed more in discussion section of the manuscript. 

Line 75-78

It is important to ensure proper referencing for the statement regarding Nepal's involvement in the mentioned initiatives, the development of action plans to reduce preventable deaths among mothers and children, and the considerable investment in strengthening information systems for performance management and service delivery. Adding appropriate references will not only provide credibility to the information but also give readers the opportunity to explore the sources and evidence supporting these claims.

Response: Authors made proper referencing for the statement regarding Nepal’s involvement in initiatives to reduce preventable deaths among mothers and children, as well as related investments in strengthening information systems.

Line 81-82

Is this claim made by the authors or by DHIS2? Proper referencing for this statement is necessary to clarify the source of the information and to ensure its accuracy and legitimacy.

Response: This is the claim based on a literature review. The authors supported this statement, ensuring accuracy and legitimacy through proper citation.

Line 84-86

To prevent redundancy, kindly integrate this sentence with lines 82-84, for example: "The Ministry of Health and Population (MoHP) in Nepal introduced DHIS2 nationally as an electronic platform for Health Management Information System (HMIS) data management since 2016. Additionally, authors are encouraged to verify whether IHIMS and HMIS refer to the same system. If IHIMS represents recent changes to HMIS, please adjust the wording accordingly. Proper referencing for this information is also essential to ensure transparency and accuracy."

Response: This comment is well taken and addressed as suggested. 

Line 105-107

Supporting reference for the statement required.

Response: Reference provided for the status of specific SDG indicators.

Line 105-111

I recommend that the authors incorporate these paragraphs into the introduction section. This is because the provided information pertains to provincial health indicators, which can be seamlessly linked with the section discussing the current gaps and the significance of the study in the introduction part. By doing so, the authors can create a cohesive and contextually relevant introduction, highlighting the importance of these indicators in framing the research objectives and addressing the study's significance.

Response: Incorporated the lines 105-111 in introduction section as suggested. 

Line 120

Kindly ensure uniformity throughout the entire manuscript by consistently using either "MCH" or "MNCH" terminology. The authors have used "MCH" in some sections and "MNCH" in others, creating inconsistency. It is essential to pick one term and apply it consistently across the entire document to maintain clarity and coherence

Response: Comment addressed. 

Line 121:

The authors mentioned that in 2021/22, DHIS2 contained a total of 12,509 reports. However, there is uncertainty regarding the definition of one report. Is it specified as one Maternal and Child Health (MCH) report submitted by one health facility on a monthly basis? To ensure clarity, the authors are requested to provide further clarification on the definition of a single report within DHIS2.

Response: Report in the manuscript refers to the monthly report submitted by each health facility (HMIS 9.3/9.4/9.5). This report includes individual sections for immunization, safe motherhood, nutrition, and IMNCI.

Line 125-126

Referencing (20) not required since its general statement.

Response: Removed the reference as suggested.

Line 164-165

Authors explain completeness was approximately 100% but they didn’t mention what was the exact figure.It is recommended to mention exact figure in scientific studies.

Response: Authors addressed the comment by mentioning exact figures.

Rather than displaying the dataset completeness results in a table, it is recommended for authors to present them using bar diagrams or other visual methods. This approach facilitates easier comparison of completeness levels across different datasets for readers.

Response: Authors presented the completeness and timeliness related data in bar diagram as recommended. 

Line 218

Please provide the full form of vaccines to make it easier to understand. For example, Penta 1 refers to the first dose of the Pentavalent Vaccine. Provide information on annex section.

Response: Addressed the comment 

Line 223

unit required for iron for e.g. Tablet/Capsule

Response: Addressed the comment

Line 255-256

The authors have highlighted that the challenges faced in achieving completeness of DHIS2 reporting are not unique and have been documented in other studies as well. It is important to delve deeper into this issue to explore whether the difficulties are attributed to end users' discomfort with the system. Additionally, it is crucial to examine whether similar challenges exist in other countries, apart from Nepal.

Response: By accepting the suggestion, the authors explored the presence of similar challenges in other countries and cited them in the manuscript. However, this study has a limitation: conducting an in-depth analysis of challenges related to the completeness of data in Nepal based on a review of DHIS2 data would not be relevant.

Line 259-260

The authors asserted that the assessment result of timeliness is higher in this province compared to other provinces of Nepal; however, no specific reference data was provided in the entire manuscript. To support this claim, it is crucial to cite the relevant sources or present evidence within the manuscript to validate the statement.

Response: Yes, result of timeliness is higher in Lumbini province compared to other provinces of Nepal as reflected in HMIS online platform (DHIS2). To support this statement, the Authors now provided the reference. 

Line 333-334

Please check the citation is well maintained as pr journal guideline.

Response: Corrected, thank you

Reviewer #6: MORE DESCRIPTION OF THE MANUFACTURING PROGRAM WILL ADD INFORMATION TO THE READER FOR PEOPLE IN OTHER COUNTRIES.

Response: By accepting the comment, authors tried to add additional information about the program as relevant. 

Reviewer #7: the author has adjusted the manuscript according to the suggestions of the previous reviewer's input and corrected the manuscript. well done

Reviewer #8: (No Response)

Reviewer #9

1. Abstract:

a) Methods section: My suggestion is to not label the study as cross-sectional. The study is ecological in nature since monthly aggregated data is utilized and there is no access to individual data. Another option is to use the term “descriptive study”.

b) Methods section: In a sentence, please mention the number of data elements selected for each objective. Readers might first assume that all 23 data items were used for all objectives, which isn’t the case.

Response: 

a) Revised the study as descriptive. 

b) Mentioned the number of data elements selected for each objective as suggested. 

2. Main Text:

a) Methods: Same comment as 1a regarding study design.

b) Line 138: Please add a comma before “and Safe Motherhood”. Otherwise, “Nutrition and Safe Motherhood” is interpreted as a single dataset.

c) For all tables, please add footnotes explaining the abbreviations used in them. Readers will find it cumbersome to search for the abbreviations in the main text. For example, in Table 1 please explain ANC, PNC, what is the meaning of ‘x’ in the column etc. Also, mention the relevant years for each table.

d) Table 2: While assessing outliers the authors have selected ±2SD. This method is apt for normally distributed data. No tests of normality or graphical visualizations were conducted and selecting ±2SD as the benchmark could be problematic. It would be better to conduct some basic tests to verify normality which isn’t complex. Additionally, please mention if selecting ±33% and ±10% as the benchmark for other metrics of consistency is specifically mentioned in the WHO data quality report card (or any other reference). If not, wjhat was the reasoning behind the criteria?

e) In the first paragraph of the results section (lines 183-184), it is mentioned that “5 out of 12 districts have 100% completeness for nutrition and IMNCI datasets (in each) and 4 out of 10 districts have 100% completeness for safe motherhood datasets”. However, in Table 3, I count 7 provinces with 100% completion for nutrition and IMNCI individually. Similarly, the results for safe motherhood do not match. Table 3 shows 8 out of 12 districts with 100% completion (and not 4 out of 10). If the total districts for safe motherhood were 10, identify the dropped districts in the table/text and provide a brief explanation as to the reason for exclusion. Please correct/verify the results.

f) While reporting timeliness the authors mention “By district, 9 out of 12 districts have more than 90% reporting rate on time for the selected datasets.” By my calculations, all districts, except Rukum East, have an average above 90%, i.e. 11 out of 12 have more than a 90% score. Please correct/verify.

g) Table 6: It wasn’t clear to me how the results reported in the final row were arrived at after calculating the total outliers in every district. Provide a brief footnote with an explanation.

h) Line 331: The sentence mentions “no one of the coverage rates are flagged” instead of “none”.

Response: Thank you for the in-depth review and comments. Authors have addressed all the comments in the revised manuscript and summary of responses are also mentioned hereunder: 

a) Addressed the comment as suggested 

b) Added comma before “and Safe Motherhood”. 

c) Mentioned the footnotes mentioning the meaning of symbols used, as well as explained the abbreviations where needed. 

d) Extreme or moderate outliers (i.e. moderate outliers if values are ± 2–3 SD from the mean or > 3.5 on modified z-score method) are identified based on the criteria set by WHO Data Quality Assurance: Framework and Metrics 2022. As suggested by reviewer, we also performed normality test that showed that the data followed symmetrical distribution and this process is now clearly mentioned in the revised manuscript. Reference for selecting ±33% and ±10% as the benchmark for other metrics of consistency is mentioned in the WHO Data Quality Assurance: Framework and Metrics 2022 and also cited in the manuscript (reference number 21).

e) Thank you for pointing the mistake. We now corrected the error with careful observation of data. 

f) Verified the data and made a minor update in the narrative, written as “By district, 9 out of 12 districts have more than a 90% reporting rate on time for each of the selected datasets.

g) Footnote with explanation of calculating Average % of moderate outliers was clearly mentioned in the revised manuscript. 

h) Corrected as suggested. 

Thank you

---

## [Editor Report · Decision Letter 2]

20 Jan 2024

Quality of Routine Health Facility Data for Monitoring Maternal, Newborn and Child Health Indicators: A Desk Review of DHIS2 Data in Lumbini Province, Nepal

PONE-D-23-07291R2

Dear Dr. Sanjel,

We’re pleased to inform you that your manuscript has been judged scientifically suitable for publication and will be formally accepted for publication once it meets all outstanding technical requirements.

Kind regards,

Kanchan Thapa, MPH, MPhil

Academic Editor

PLOS ONE

Additional Editor Comments (optional):

Dear All Authors,

Thank you for revising your paper based on a series of peer reviews. I now believe the paper is worthy of publication. The paper highlights the significant use of DHIS-2 data, which has been collected over the years and is limited in current literature. It serves as an example of how accumulated data over the years can be utilized for scientific publications.

Please review my additional comments regarding authorship and formatting.

Comments on Authorship: Please ensure that all the authors meet the criteria for authorship and revise your author's contribution section accordingly. For detailed information, refer to the following link: https://journals.plos.org/plosone/s/authorship. If any of the authors do not meet the above criteria, please acknowledge them in the acknowledgment section.

Issues on Formatting: While reading your paper, on line number 200, I found a statement that reads, "Key dimensions included for assessing internal consistency include:…". Please clearly indicate what the key dimensions are, or this can be resolved during the final stage of formatting. Therefore, I request you to correct this issue at this stage to ensure accuracy up to the final publication. Also, ensure all the table, figures and referencing are as per the PLOS One guidelines. 

At this stage, I would like to thank all the reviewers for their wonderful contribution to review this paper. 
---

## [Editor Report · Acceptance letter]

21 Mar 2024

PONE-D-23-07291R2 

PLOS ONE

Dear Dr. Sanjel, 

I'm pleased to inform you that your manuscript has been deemed suitable for publication in PLOS ONE. Congratulations! Your manuscript is now being handed over to our production team.

Kind regards, 

on behalf of

Mr. Kanchan Thapa 

Academic Editor

PLOS ONE